# Smoothness Bridges Sparsity and Stability in MoEs

## Abstract

Mixture-of-experts (MoE) architectures have recently emerged as an effective approach for scaling model capacity while managing computational costs by leveraging expert sparsity, where only a subset of experts is activated during inference. Despite their computational efficiency, MoE models face challenges in training stability compared to their dense counterparts, largely due to the introduction of expert sparsity. While several methods have been proposed to mitigate this instability, the underlying relationship between expert sparsity and training stability remains unclear. In this work, we develop a theoretical framework that demonstrates an inverse correlation between training stability and expert sparsity, with gradient smoothness serving as the bridge. We derive an upper bound on training stability, formalizing for the first time the sparsity-stability trade-off in MoE models. Our findings show that activating more experts enhances gradient smoothness and improves training stability but at the cost of reduced sparsity. We validate our theory through extensive experiments on various architectures and datasets, and propose a novel MoE structure that addresses stability without sacrificing sparsity. This design introduces independent router heads and a soft top-$K$ selection via sampling without replacement, which smooths the gradient landscape while maintaining expert sparsity. Further analysis confirms the promise of this structure in striking the optimal balance between sparsity and stability, offering a new direction for optimizing MoE architectures in large-scale models.

## 1 Introduction

Mixture of Experts (MoE) was introduced to implement conditional computation within a model to enhance training and inference speeds (Jacobs et al., 1991; Jordan & Jacobs, 1994). MoE was later extended to deep learning architectures, including CNNs, RNNs, and transformers (Eigen et al., 2013; Shazeer et al., 2017). MoE has since evolved to support large-scale models, as demonstrated by recent works (Jiang et al., 2024; Wei et al., 2024; Krajewski et al., 2024), which highlight the scalability advantages of sparse MoEs over dense models. For a comprehensive overview, we refer readers to surveys (Yuksel et al., 2012; Masoudnia & Ebrahimpour, 2014).

Despite the empirical success of MoEs, a critical challenge persists: the trade-off between sparsity and stability. Sparser MoEs, of which routers select fewer experts for each input, are more efficient to run but less stable to train (Shazeer et al., 2017; Fedus et al., 2022; Zoph et al., 2022). This dilemma presents challenges for MoE networks in terms of overfitting and fine-tuning. To address the instability of MoE

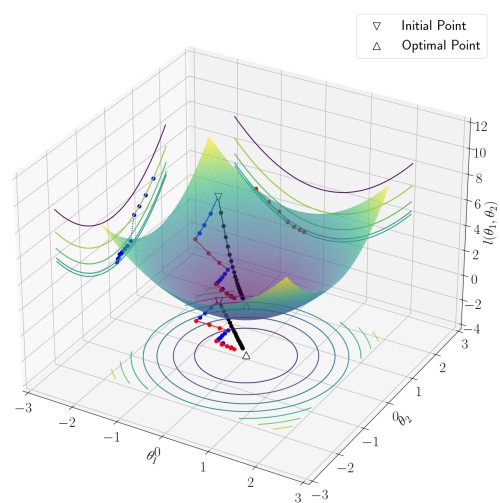

Figure 1: **An illustration of sparsity-stability trade-off in MoEs**.

training, researchers have explored various strategies, such as, stochastic or differentiable top-$K$ selection for soft routing (Xie & Ermon, 2019; Paulus et al., 2020; Hazimeh et al., 2021), and different dense training strategies bypassing sparse routing (Nie et al., 2021; Komatsuzaki et al., 2022; Chen et al., 2023; Pan et al., 2024). However, the exact relationship between expert sparsity and training stability remains unclear, hindering theoretical guidance for improving MoE training and limiting their broader application.

We begin with an illustrative example of MoE training (see Figure 1 and Appendix A for details). In Figure 1, the $x$-axis ($\theta_1$) represents expert 1's parameter, the $y$-axis ($\theta_2$) represents expert 2's parameter, and the $z$-axis ($l(\theta_1, \theta_2)$) denotes the loss. The 3D surface illustrates the loss landscape, while the 2D contours show its projections. The blue-red lines (top-1 optimization in the full space) exhibit far more zigzagging compared to the black lines (top-2 optimization in the full space), although the projected blue and red lines (top-1 optimizations in sub-spaces) remain relatively smooth. This suggests that the loss landscape of a dense MoE is smoother than that of a sparse MoE, leading to more predictable gradients and stable training. This insight leads us to the following question:

***Does selecting more experts in MoEs result in smoother optimization and more stable training?***

In this work we investigate the above question through both theoretical and empirical investigations, where our main contributions can be summarized as follows:

- **We propose a theoretical framework connecting expert sparsity and training stability via gradient smoothness.** By introducing the concept of gradient smoothness and analyzing the Lipschitz constants of both the loss function and its gradient, we establish the quantitative relationship between sparsity and stability. We obtain the theoretical upper bound on training stability for MoEs, formulating the sparsity-stability trade-off in MoE models for the first time. Our theory shows that activating more experts enhances gradient smoothness and improves training stability upper bound, however at the cost of reduced sparsity.

- **Our experimental investigation across representative architectures verify the universal efficacy of our theory.** Specifically, we perform extensive experiments using MoE architectures based on Multi-Layer Perceptrons (MLPs), Convolutional Neural Networks (CNNs), and transformers, applied to synthetic, image, and text datasets. These experiments consistently demonstrate that denser MoEs result in smoother gradients and more stable training across various architectures and datasets, supporting our theoretical findings.

- **We introduce a novel MoE structure that improves stability without sacrificing sparsity, striking the sparsity-stability trade-off.** Under the guidance of our theory, we address two key issues in conventional MoE models: zero gradients in top-1 MoEs and the deterministic nature of top-$K$ selection. Our design introduces independent router heads and uses a soft top-$K$ selection through sampling without replacement, smoothing the gradient landscape while maintaining expert sparsity. This structure achieves stability bounds comparable to dense models, offering a solution to the sparsity-stability trade-off in MoEs.

This paper is structured as follows: Section 2 covers related works, followed by the preliminaries in Section 3. In Section 4, we present our theoretical findings and introduce our new MoE structure. Section 5 details our empirical results. Section 6 concludes our paper.

## 2 RELATED WORKS

**Training Instability of MoEs**: Compared to dense networks, MoEs have been noted for their poorer stability and generalization, as highlighted by (Shazeer et al., 2017; Fedus et al., 2022; Zoph et al., 2022). These issues make MoEs prone to overfitting and challenging to fine-tune. Foundational works by Shazeer et al. (Shazeer et al., 2017) and subsequent studies by Fedus et al. (Fedus et al., 2022) and Zoph et al. (Zoph et al., 2022) identified these challenges and initiated the exploration of methods to mitigate them. To address the instability of MoEs, researchers have explored stochastic or differentiable top-$K$ selection for soft routing (Xie & Ermon, 2019; Paulus et al., 2020; Hazimeh et al., 2021) and different dense training strategies bypassing sparse routing (Nie et al., 2021; Komatsuzaki et al., 2022; Chen et al., 2023; Pan et al., 2024). These studies provided critical insights

into the instability problem but did not fully address the connection between expert sparsity and training stability, which our work aims to explore further.

**Gradient Smoothness and Training Stability**: Mini-batch Stochastic Gradient Descent (SGD) (Robbins & Monro, 1951) is a widely utilized optimization method in deep learning. The convergence properties of mini-batch SGD are well-established under certain smoothness and convexity conditions (Garrigos & Gower, 2023). Recent studies have extensively examined the critical role of gradient smoothness in achieving stable and generalizable mini-batch SGD methods (Hardt et al., 2016; Charles & Papailiopoulos, 2018; Kuzborskij & Lampert, 2018; Wu et al., 2018; Lei & Ying, 2020). Techniques such as weight decay (Krogh & Hertz, 1991), gradient clipping (Mikolov et al., 2012), network pruning (Srivastava et al., 2014), and batch normalization (Ioffe & Szegedy, 2015) have been proposed to enhance the stability of mini-batch SGD. These foundational studies on gradient smoothness and stability directly inform our work, as we extend these concepts to the sparsity of MoEs. By connecting gradient smoothness with the expert selection process in MoEs, our work builds upon these established principles to propose a theoretical framework that links expert sparsity with training stability.

## 3 PRELIMINARIES

In this section, we introduce the foundational concepts and notations that will be used throughout this paper, focusing on the structure and properties of MoE networks and the mathematical definitions related to smoothness and stability of the training process.

### 3.1 MIXTURE OF EXPERTS STRUCTURE

We consider a MoE network $\mathcal{F}$ composed of $N$ MoE blocks, expressed as:

$$\mathcal{F} = \mathcal{F}_1 \circ \mathcal{F}_2 \circ \cdots \circ \mathcal{F}_N.$$

The output of each block $\mathcal{F}_i$ is a weighted average of the selected $K_i$ experts out of the total $M_i$ experts in $\mathcal{F}_i$:

$$\mathcal{F}_i\left(\boldsymbol{\Theta}_i; \mathbf{x}\right) = \sum_{j \in \mathcal{T}_i} \mathcal{G}_{i,j}\left(\boldsymbol{\Theta}_i^g; \mathbf{x}\right) f_{i,j}\left(\boldsymbol{\Theta}_{i,j}^f; \mathbf{x}\right).$$

Here, $i$ is the block index, and $j$ is the expert index. $\mathbf{x}$ denotes the inputs to the whole network $\mathcal{F}$. We also denote $\mathbf{y}$ as the output of the whole network $\mathcal{F}$, and $\mathbf{z}_i$ as the input of the MoE block $\mathcal{F}_i$ for $i = 1, 2, \ldots, N$ with $\mathbf{z}_1 = \mathbf{x}$. The set $\mathcal{T}_i$ represents the indices of selected experts in block $\mathcal{F}_i$. The function $f_{i,j}$ denotes the $j$-th expert in $\mathcal{F}_i$. The function $\mathcal{G}_{i,j}$ represents the router probability head for $f_{i,j}$, outputting a probability value for MoE averaging. The parameters of block $\mathcal{F}_i$ are denoted by $\boldsymbol{\Theta}_i = \left[\boldsymbol{\Theta}_i^g, \boldsymbol{\Theta}_{i,1}^f, \ldots, \boldsymbol{\Theta}_{i,M_i}^f\right]$, where $\boldsymbol{\Theta}_i^g$ and $\boldsymbol{\Theta}_{i,j}^f$ are the parameters of the router $\mathcal{G}_i$ and expert $f_{i,j}$ in block $\mathcal{F}_i$. The parameters of network $\mathcal{F}$ is denoted by $\boldsymbol{\Theta} = [\boldsymbol{\Theta}_1, \ldots, \boldsymbol{\Theta}_N]$.

The output of router probability head $\mathcal{G}_{i,j}$ is computed as:

$$\mathcal{G}_{i,j}\left(\boldsymbol{\Theta}_i^g; \mathbf{x}\right) = \operatorname{softmax}\left(\operatorname{TopK}_i\left(g_{i,j}(\boldsymbol{\Theta}_i^g; \mathbf{x})\right)\right)$$

$$= \frac{\mathbf{1}_{j \in \mathcal{T}_i} \cdot \exp\left(g_{i,j}(\boldsymbol{\Theta}_i^g; \mathbf{x})\right)}{\sum_{k \in \mathcal{T}_i} \exp\left(g_{i,k}(\boldsymbol{\Theta}_i^g; \mathbf{x})\right)}.$$

Here, $\mathbf{1}_{j \in \mathcal{T}_i}$ is the indicator function, which outputs 1 if $j \in \mathcal{T}_i$ and 0 otherwise. And $g_{i,j}$ denotes the router value head for $f_{i,j}$, typically implemented as a MLP network. The most common method for constructing $\mathcal{T}_i$ is by top-$K_i$ sorting, where experts $f_{i,j}$ are sorted by their corresponding router values $g_{i,j}$, and the first $K_i$ experts are selected and weighted by their associated router probabilities to compute the output of the MoE block $\mathcal{F}_i$.

### 3.2 SMOOTHNESS OF LOSS FUNCTION

Smoothness of a loss function is an important factor that impacts the performance of gradient descent methods. A smooth loss function typically has a gradient that varies gradually with changes in the parameters, leading to a more stable convergence behavior during the optimization process. In

contrast, a non-smooth or highly irregular loss function can result in a noisier gradient estimate, which may slow down the convergence of gradient descent. Here, we outline the key mathematical concepts pertaining to the smoothness of loss functions.

A loss function $\mathcal{L} : \Omega \times \mathbb{X} \to \mathbb{R}$ is convex if, for all $\boldsymbol{\Theta}, \boldsymbol{\Theta}' \in \Omega$, the following holds:

$$\mathcal{L}\left(\boldsymbol{\Theta}'; \mathbf{x}\right) \geq \mathcal{L}\left(\boldsymbol{\Theta}; \mathbf{x}\right) + \nabla_{\boldsymbol{\Theta}} \mathcal{L}\left(\boldsymbol{\Theta}; \mathbf{x}\right)^{\top} \left(\boldsymbol{\Theta}' - \boldsymbol{\Theta}\right).$$

And a loss function $\mathcal{L}$ is $L$-Lipschitz if, for all $\boldsymbol{\Theta} \in \Omega$ and $\mathbf{x} \in \mathbb{X}$, the gradient satisfies $\|\nabla_{\boldsymbol{\Theta}} \mathcal{L}\left(\boldsymbol{\Theta}; \mathbf{x}\right)\|_2 \leq L$, i.e., $L$ is an upper bound on the 2-norm of the loss gradients. This implies:

$$\|\mathcal{L}\left(\boldsymbol{\Theta}; \mathbf{x}\right) - \mathcal{L}\left(\boldsymbol{\Theta}'; \mathbf{x}\right)\| \leq L \|\boldsymbol{\Theta} - \boldsymbol{\Theta}'\|.$$

Similarly, $\mathcal{L}$ is $\beta$-smooth if its gradient $\nabla_{\boldsymbol{\Theta}} \mathcal{L}$ is $\beta$-Lipschitz, meaning:

$$\|\nabla_{\boldsymbol{\Theta}} \mathcal{L}\left(\boldsymbol{\Theta}; \mathbf{x}\right) - \nabla_{\boldsymbol{\Theta}'} \mathcal{L}\left(\boldsymbol{\Theta}'; \mathbf{x}\right)\| \leq \beta \|\boldsymbol{\Theta} - \boldsymbol{\Theta}'\|.$$

### 3.3 STABILITY OF TRAINING METHOD

In this work, stability refers to the sensitivity of an optimization algorithm to changes in the training data. A stable algorithm produces similar outputs when the data is slightly altered, often leading to better generalization.

We use mini-batch Stochastic Gradient Descent (SGD) to train MoEs:

$$\boldsymbol{\Theta}_{t+1} := \mathcal{U}\left(\boldsymbol{\Theta}_t; \mathcal{B}\right) = \boldsymbol{\Theta}_t - \alpha_t \frac{\sum_{\mathbf{x}_i \in \mathcal{B}} \nabla_{\boldsymbol{\Theta}} \mathcal{L}\left(\boldsymbol{\Theta}_t; \mathbf{x}_i\right)}{B},$$

where $t$ is the iteration, $\mathcal{B}$ is the batch, $B$ is the batch size, and $\alpha_t$ is the learning rate. We omit tuples $(\boldsymbol{\Theta}; \mathbf{x})$ when clear from context.

An SGD update rule $\mathcal{U}$ is $\epsilon$-uniformly stable (Hardt et al., 2016) if, for datasets $\mathcal{B}, \mathcal{B}' \in \mathbb{X}^{|\mathcal{B}|}$ differing by one point, the following holds:

$$\sup_{\mathbf{x}} \mathbb{E}_{\mathcal{U}} \left[\|\mathcal{L}(\mathcal{U}(\mathcal{B}); \mathbf{x}) - \mathcal{L}\left(\mathcal{U}\left(\mathcal{B}'\right); \mathbf{x}\right)\|_2\right] \leq \epsilon. \tag{1}$$

Here, the expectation is over the internal stochasticity of mini-batch selection. We denote by $\epsilon_{\text{stab}}(\mathcal{U}, |\mathcal{B}|)$ the infimum of $\epsilon$ for which this holds, with lower values indicating better stability.

In (Hardt et al., 2016), an upper bound on $\epsilon_{\text{stab}}$ of the form $O\left(L^2 \alpha_t\right)$ is derived. Given that SGD converges for convex loss if and only if $\alpha_t \leq \frac{1}{4\beta}$ (Garrigos & Gower, 2023), $\epsilon_{\text{stab}}$ is consequently bounded by $O\left(\frac{L^2}{\beta}\right)$ with a convergence guarantee. Inspired by this result, we adopt $\frac{L^2}{\beta}$ as a measure of gradient smoothness, as it is closely tied to $\epsilon_{\text{stab}}$.

## 4 THEORETICAL ANALYSIS

In this section, we present the main results of our theoretical analysis on the relationship between expert sparsity, gradient smoothness, and training stability in MoE models. Detailed proofs can be found in Appendix B and C.

We define key terms as follows: (1) *Expert sparsity*, quantified by the top-$K$ parameter, refers to the number of activated experts; (2) *Training stability*, formalized by $\epsilon_{\text{stab}}$, measures the stability of mini-batch SGD updates; (3) *Gradient smoothness*, evaluated by $\frac{L^2}{\beta}$, reflects how rapidly the loss function can change. Sparsity $K$ ranges from 1 to the total number of experts in the MoE block, with smaller $K$ indicating greater sparsity. Both $\epsilon_{\text{stab}}$ and $\frac{L^2}{\beta}$ range from 0 to infinity, with higher values implying less stability and smoothness in MoEs.

The following assumption on the local properties of the MoE loss function is essential for all theoretical results in this section:

**Assumption 1 (*Local properties of the MoE loss*).** *Assume the loss function $\mathcal{L}(\boldsymbol{\Theta}; \mathbf{x})$ is locally convex, $\beta$-smooth, and $L$-Lipschitz for every $(\boldsymbol{\Theta}, \mathbf{x}) \in \mathcal{B}_\epsilon(\tilde{\boldsymbol{\Theta}}) \times \mathbb{X}$, where $\mathcal{B}_\epsilon(\tilde{\boldsymbol{\Theta}}) := \left\{\boldsymbol{\Theta} \mid \|\boldsymbol{\Theta} - \tilde{\boldsymbol{\Theta}}\|_2 \leq \epsilon, \boldsymbol{\Theta}, \tilde{\boldsymbol{\Theta}} \in \Omega\right\}$ denotes the neighborhood of $\tilde{\boldsymbol{\Theta}}$ in $\Omega$.*

## 4.1 SPARSITY-STABILITY TRADE-OFF IN MoEs

With the key terms and assumptions established, we now analyze specific properties of the MoE model under different configurations. Starting with the case where top-$K_i = 1$, we observe notable behavior in the router gradients.

**Proposition 1** *(Zero gradients for top-1 MoEs). For $i = 1, 2, \ldots, N$, if $K_i = 1$, the Jacobian of the router in MoE block $i$ is zero, i.e., $\nabla_{\boldsymbol{\Theta}_i^g} \mathcal{L} = 0$.*

Proposition 1 reveals that selecting a single expert (top-$K_i = 1$) leads to vanishing router gradients, hindering effective routing. For top-$K_i > 1$, we derive the following lemma on the trade-off between sparsity $K$ and $L$-Lipschitzness and $\beta$-smoothness.

**Lemma 1** *(Sparsity-smoothness trade-off in MoEs). Under Assumption 1, with top-$K_i = K > 1$ for $i = 1, 2, \ldots, N$, the L-Lipschitz constant of the MoE loss function is:*

$$L = O\left(\frac{1}{\sqrt{K}}\right),$$

*and the $\beta$-smoothness constant is:*

$$\beta = O\left(\frac{1}{\sqrt{K}}\right).$$

Lemma 1 establishes the foundation for linking sparsity and stability in MoEs (see Appendix B). Let $\epsilon_{\text{stab}}^{\text{MoE}}$ denote the stability of the MoE, defined as in Equation 1. The following theorem formalizes the trade-off between stability $\epsilon_{\text{stab}}^{\text{MoE}}$ and sparsity $K$.

**Theorem 1** *(Sparsity-stability trade-off in MoEs). Under Assumption 1, with $K_1 = K_2 = \cdots = K_N = K \geq 1$, and mini-batch SGD with fixed step size $\alpha = 1/(4\beta)$ for $T$ steps in $\mathcal{B}_\epsilon(\tilde{\boldsymbol{\Theta}})$, the MoE achieves uniform stability:*

$$\epsilon_{\text{stab}}^{\text{MoE}} \leq O\left(\frac{T}{\sqrt{K}B}\right).$$

The theorem shows that stability is inversely related to top-$K$, the number of activated experts. Increasing $K$, activating more experts, improves stability at the expense of sparsity. While decreasing $K$, activating less experts, reduces stability. It underscores the need to balance expert selection and stability in MoEs.

## 4.2 DESIGNING MORE STABLE MoEs

Proposition 1 and Theorem 1 highlight two key issues with standard MoEs: (1) zero router gradients for top-1 MoEs (see Appendix B for more details), and (2) instability in sparse MoEs (small top-$K$). The former is due to shared router parameters, while the latter stems from deterministic sorting in top-$K$.

To address these, we propose the following modification:

$$\tilde{\mathcal{G}}_{i,j}\left(\boldsymbol{\Theta}_{i,j}^g; \mathbf{x}\right) = \frac{\mathbf{1}_{j \in \tilde{\mathcal{T}}_i} \cdot \exp\left(\tilde{g}_{i,j}\left(\boldsymbol{\Theta}_{i,j}^g; \mathbf{x}\right)\right)}{\sum_{k \in \tilde{\mathcal{T}}_i} \exp\left(\tilde{g}_{i,k}\left(\boldsymbol{\Theta}_{i,j}^g; \mathbf{x}\right)\right)},$$

where $\tilde{\mathcal{G}}$ incorporates (Figure 2):

- **Multi-headed routing**: Each output $\tilde{g}_{i,j}$ is generated by an independent network with parameter $\boldsymbol{\Theta}_{i,j}^g$, unlike single-headed routers with shared parameters (Figure 2(b)).

- **Soft top-$K$ by sampling**: Stochastic top-$K$ is implemented via Gumbel-softmax sampling, without replacement, from the distribution $p_j = \text{softmax}(\tilde{g}_{i,j})$, generating the sampled indices set $\tilde{\mathcal{T}}_i$ (Figure 2(d)).

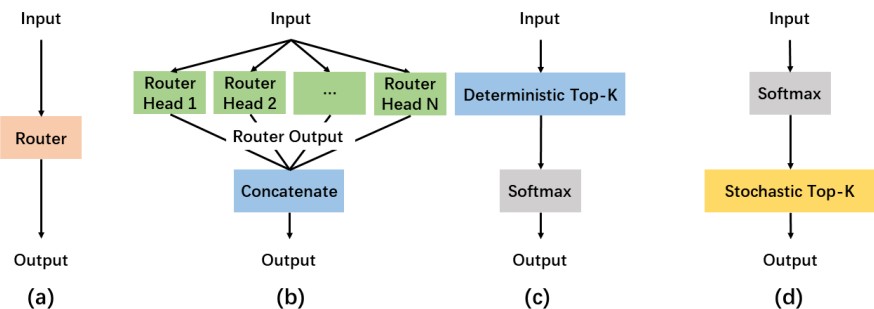

Figure 2: **MoE architecture comparison.** (a) Single-headed routing. (b) Multi-headed routing. (c) Deterministic top-$K$. (d) Soft top-$K$ by sampling.

The experts $f_{i,j}$ remain unchanged, and the output of the new MoE structure is:

$$\tilde{\mathcal{F}}_i\left(\boldsymbol{\Theta};\mathbf{x}\right) = \sum_{j\in\tilde{\mathcal{T}}_i}\tilde{\mathcal{G}}_{i,j}\left(\boldsymbol{\Theta}_{i,j}^g;\mathbf{x}\right)f_{i,j}\left(\boldsymbol{\Theta}_{i,j}^f;\mathbf{x}\right).$$

We refer to this structure as soft multi-headed MoE and denote its stability as $\epsilon_{\text{stab}}^{\text{mod-MoE}}$. The following theorem establishes the stability of the modified MoE:

**Theorem 2** (*Stability of modified MoE*) *Under Assumption 1, with $M_1 = M_2 = \cdots = M_N = M \geq 1$, and running mini-batch SGD with fixed step size $\alpha = 1/(4\beta)$ for $T$ steps in $\mathcal{B}_\epsilon(\tilde{\boldsymbol{\Theta}})$, the stability bound is:*

$$\epsilon_{\text{stab}}^{\text{mod-MoE}} \leq O\left(\frac{T}{\sqrt{M}B}\right).$$

Recall that $M$ is the total number of experts in each MoE block, with $M \geq K$, the number of activated experts. Theorem 2 shows that the modified MoE $\tilde{\mathcal{F}}$ pushes the stability bound in Theorem 1 to its theoretical extreme value reached when all the $M$ experts are activated, by activating only $K$ of them. Therefore, the modified MoE $\tilde{\mathcal{F}}$ matches the stability bound of dense models, regardless of top-$K$, while avoiding the zero router gradient issue highlighted in Proposition 1. This suggests that the proposed model addresses the sparsity-stability trade-off. Proof details are in Appendix C, with empirical results in Section 5.

## 5 EXPERIMENTAL VERIFICATION

In Section 4, we theoretically establish a connection between expert sparsity and training stability via gradient smoothness. Based on the theory, we also propose a novel MoE structure and demonstrate its superiority. To empirically validate these theoretical findings, we conducted a comprehensive series of experiments across various models and datasets. This section details our experimental setup, implementation, evaluation metrics, and results. Through these experiments, we systematically investigate the interplay between stability, smoothness, and sparsity, providing empirical evidence to support our theoretical insights and demonstrating the practical implications for optimizing MoE architectures.

### 5.1 SETUPS

#### 5.1.1 EXPERIMENT OF MLP-MOES ON SYNTHETIC DATA

**Data**: We generate 10 categories of 128-dimensional data by sampling from Gaussian distributions with varying means and standard deviations. For each category, 2,000 data points are prepared for training, and 800 for testing, resulting in a training set size of 20,000 and a test set size of 8,000.

**Model**: Our MoE architecture consists of 1 router and 5 experts. Both the router and the experts are implemented as two-layer Multi-Layer Perceptron (MLP) networks. Each expert network is pre-trained to specialize in 2 out of the 10 data categories.

### 5.1.2 EXPERIMENT OF CNN-MoEs ON IMAGE DATA

**Data**: Our CNN model is trained on the Fashion-MNIST (Xiao et al., 2017) dataset, a ten-class classification task. The dataset comprises 60,000 training samples and 10,000 test samples.

**Model**: The MoE model used in this experiment comprises a two convolutional layers followed by a MoE block which includes 1 router and 5 experts. Similar to the synthetic-data experiments, both the router and the experts are two-layer MLP networks, with each expert pre-trained to specialize in 2 out of the 10 data categories.

### 5.1.3 EXPERIMENT OF TRANSFORMER-MoEs ON TEXT DATA

**Data**: The Transformer model is trained on the Banking77 dataset, which is designed for the task of dialogue intent prediction. The dataset contains 77 distinct intent categories, with 10,003 training samples and 3,080 test samples.

**Model**: We use the Bidirectional Encoder Representations from Transformers (BERT) (Devlin et al., 2018) model, initializing it with pre-trained parameters specifically tailored for the Banking77 dataset, followed by fine-tuning. In our MoE architecture, the feed-forward network layer is configured with 7 experts, each of which is a two-layer neural network with cubic activation.

### 5.2 IMPLEMENTATIONS

All of our models are implemented using PyTorch, and we utilize the mini-batch SGD optimizer for the training process. The MLP-MoE and CNN-MoE models are trained for 100 epochs with a learning rate of 0.001 and a batch size of 256. The Transformer model is trained for 15 epochs with a learning rate of 0.00005 and a batch size of 64. Model parameters are recorded at each epoch for downstream analysis of sparsity, smoothness and stability.

### 5.3 EVALUATIONS

We evaluate training stability $\epsilon_{\text{stab}}$, $L$-Lipschitzness, and $\beta$-smoothness using the finite difference method. At a certain step during the training process, we randomly select three data batches of 64 samples from the training set, denoted as $\mathcal{B}_1$, $\mathcal{B}_2$, and $\mathcal{B}_3$. To explore the neighborhood of $\Theta_0$, the parameters of the current network, we use respectively $\mathcal{B}_1$ and $\mathcal{B}_2$ to update the network, resulting in two independently updated parameters $\Theta_1$ and $\Theta_2$, as well as the corresponding Jacobians $\nabla_{\Theta_1}\mathcal{L}$ and $\nabla_{\Theta_2}\mathcal{L}$. The values of $\epsilon_{\text{stab}}$, $L$, and $\beta$ at the current step are then evaluated over $\mathcal{B}_3$ using finite differences: $\epsilon_{\text{stab}}(\mathcal{B}_1, \mathcal{B}_2) = \frac{\sum_{\mathbf{x} \in \mathcal{B}_3} \|\mathcal{L}(\Theta_1; \mathbf{x}) - \mathcal{L}(\Theta_2; \mathbf{x})\|_2}{64}$, $L(\mathcal{B}_1, \mathcal{B}_2) = \frac{\sum_{\mathbf{x} \in \mathcal{B}_3} \|\mathcal{L}(\Theta_1; \mathbf{x}) - \mathcal{L}(\Theta_2; \mathbf{x})\|_2}{64\|\Theta_1 - \Theta_2\|_2}$, and $\beta(\mathcal{B}_1, \mathcal{B}_2) = \frac{\sum_{\mathbf{x} \in \mathcal{B}_3} \|\nabla_{\Theta_1}\mathcal{L}(\Theta_1; \mathbf{x}) - \nabla_{\Theta_2}\mathcal{L}(\Theta_2; \mathbf{x})\|_2}{64\|\Theta_1 - \Theta_2\|_2}$.

Considering the impact of stochasticity in mini-batch SGD, we repeat the above experiment 200 times with different data batches $(\mathcal{B}_1, \mathcal{B}_2, \mathcal{B}_3)$. The largest values of $\epsilon_{\text{stab}}$, $L$, $\beta$ obtained are taken as the evaluation estimates for the current training step, denoted as $\tilde{\epsilon}_{\text{stab}}$, $\tilde{L}$, and $\tilde{\beta}$, respectively. Gradient smoothness is then computed as $\frac{\tilde{L}^2}{\tilde{\beta}}$.

Finally, we compute the average of the measurements taken across five trained models in the replicated experiments to determine our final values for smoothness and stability.

### 5.4 RESULTS

### 5.4.1 IMPACT OF EXPERT SPARSITY ON CONVERGENCE AND LOSS VARIANCE

To demonstrate the impact of expert sparsity on training stability as indicated by our theoretical analysis, we investigate the training loss and the variance of loss over the training process. Using a sliding window method (with a window size of 101), we calculate the variance of loss within each window as our measure of loss variance. As shown in Figure 3, the sparsest (top-1) network converges more slowly and has a higher loss variance compared to the densest (top-5) network, consistent with our theory.

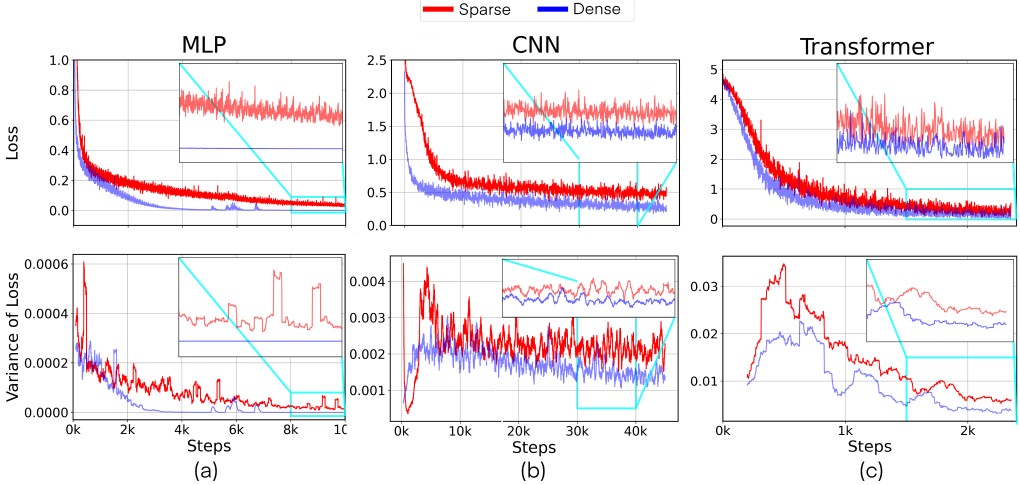

Figure 3: **Impact of Expert Sparsity on Convergence and Loss Variance.** (a) MLP. (b) CNN. (c) Transformer. (Top) The sparse network (red line, top-$K = 1$) exhibits slower convergence compared to the dense network (blue line, top-$K = 5$). (Bottom) The sparse network shows greater variance in loss during training, indicating a less stable training process relative to the dense network.

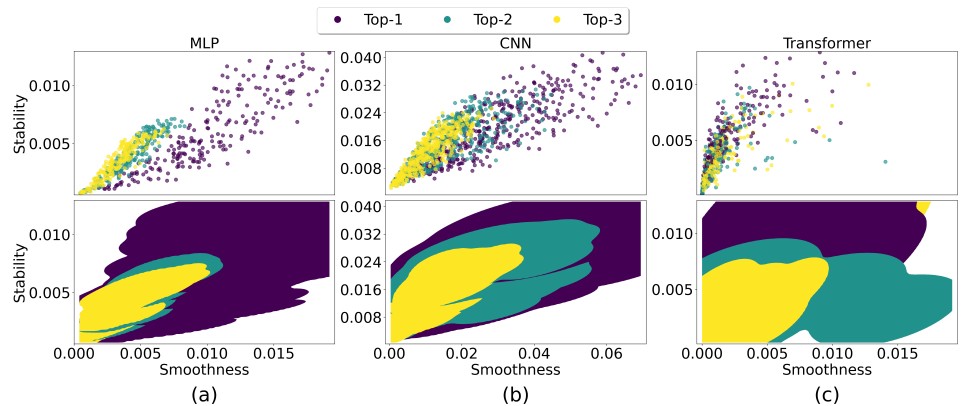

Figure 4: **Correlation Between Smoothness and Stability across Top-$K$.** (a) MLP. (b) CNN. (c) Transformer. (Top) Each data point represents the gradient smoothness ($\frac{L^2}{\beta}$) and training stability ($\epsilon_{\text{stab}}^{\text{MoE}}$) measured from a MoE network, with different colors indicating different top-$K$ values. (Bottom) The spread of the smoothness-stability region for different top-$K$ settings is illustrated with corresponding colors.

### 5.4.2 CORRELATION BETWEEN SMOOTHNESS AND STABILITY ACROSS TOP-$K$

Similarly, to illustrate the relationship between gradient smoothness and training stability, we examine both the smoothness ($\frac{L^2}{\beta}$) and stability ($\epsilon$) at different steps during the model training process. As shown in the top row of Figure 4, each point denotes a measurement of smoothness and stability for a model at a certain training step in an experiment. It can be seen that the sparsest network (top-1) exhibits the widest dispersion in the upper right, indicating the worst smoothness and stability. In contrast, denser networks tend to concentrate in the lower left, exhibiting better smoothness and stability. Bottom row of Figure 4 exhibits the distribution area for varying top-$K$ by stacking thresholded density plot of the scattered points. As top-$K$ increases from 1 to 5, an obvious trend of distribution shifting from the upper right to the lower left is present. It demonstrates a negative relationship between sparsity and both stability and smoothness.

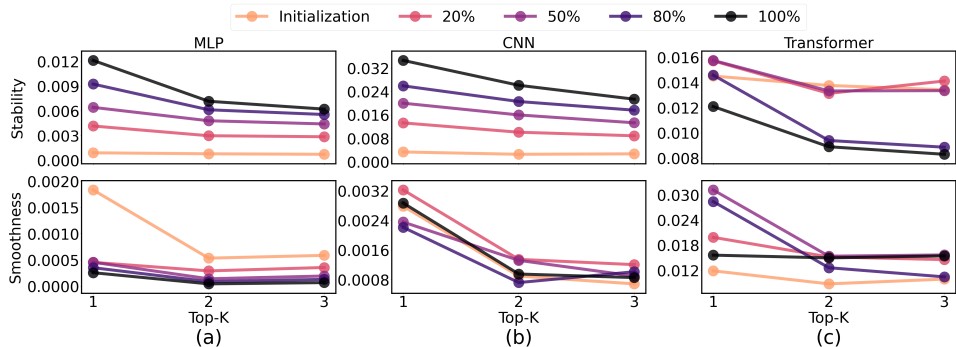

Figure 5: **Evolution of Stability and Smoothness Throughout Training Stages.** (a) MLP. (b) CNN. (c) Transformer. Each subplot presents the gradient smoothness $\frac{L^2}{\beta}$ (upper panel) and training stability $\epsilon_{\text{stab}}$ (lower panel) across different top-$K$ settings. The x-axis represents top-$K$, while the y-axes indicate stability (top) and smoothness (bottom). Each point is an average of the values in the corresponding stage. Color shading illustrates the progression of training stages, with darker shades corresponding to later stages. The figure highlights how different top-$K$ values influence the model's smoothness and stability throughout the training process, with denser settings showing more concentrated and stable behaviors.

### 5.4.3 EVOLUTION OF STABILITY AND SMOOTHNESS THROUGHOUT TRAINING STAGES

To further investigate the impact of expert sparsity on training stability and gradient smoothness, we conduct experiments at different training stages for networks trained with varying top-$K$ values. By adjusting the top-$K$ hyperparameter, we measure the resulting smoothness and stability. As shown in Figure 5, stability and smoothness exhibit distinct trends with changes in top-$K$: during the early stages of training, differences among top-$K$ values (except for top-1) are minimal, with high-sparsity networks even showing lower initial smoothness. However, as training progresses, the gap in smoothness and stability between networks of different sparsities becomes more pronounced. Notably, the network trained with top-1 behaves differently from the others, consistent with our theoretical observation that a top-1 router is nearly inactive, potentially explaining its distinct behavior. As training advances, the instability and lack of smoothness in sparse networks become more apparent, consistent with the inverse relationship between sparsity and both stability and smoothness.

### 5.4.4 IMPROVED STABILITY FOR SOFT TOP-$K$ MULTI-HEADED MOES

To validate the effectiveness of the proposed soft top-$K$ multi-headed MoE structure, we conduct experiments across different architectures, including MLP, CNN, and Transformer models, to analyze its stability and performance relative to traditional deterministic top-$K$ MoEs.

As shown in Figure 6, the single-headed MoE with deterministic routing (orange lines) frequently suffers from vanishing gradients, particularly in sparse settings (top-$K = 1$), leading to training instability. In contrast, our proposed multi-headed MoE (blue lines) consistently maintains non-zero gradients throughout training, ensuring smoother and more stable convergence. This demonstrates that the multi-headed routing strategy effectively addresses the zero-gradient issue inherent in top-1 MoEs, particularly in scenarios where sparse selection of experts is critical.

Additionally, as depicted in Figure 7, we compare the training stability $\epsilon_{\text{stab}}$ across three routing strategies: deterministic top-$K$ followed by softmax (red lines), softmax followed by deterministic top-$K$ (orange lines), and softmax followed by stochastic top-$K$ via sampling without replacement (blue lines). The results indicate that the softmax followed by stochastic top-$K$ routing provides the highest stability (blue lines), followed by softmax with deterministic top-$K$ (orange lines), while the original deterministic top-$K$ (red lines) shows the least stability. This confirms that incorporating stochastic elements into the routing process leads to a significant improvement in training stability, particularly in sparse configurations.

In summary, these experiments demonstrate that our soft top-$K$ multi-headed MoE structure not only mitigates the gradient vanishing problem but also enhances training stability by introducing

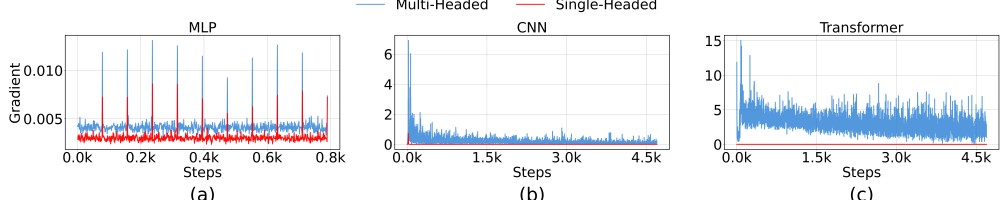

Figure 6: **Multi-Headed MoEs Eliminate Zero Router Gradients.** (a) MLP. (b) CNN. (c) Transformer. The standard MoE with a single router head (red lines) shows vanishing gradients, particularly in sparse settings (top-$K = 1$), leading to unstable training dynamics. In contrast, the proposed multi-headed MoE (blue lines) maintains non-zero gradients throughout the training process, resulting in smoother and more stable convergence. This suggests that multi-headed routing effectively addresses the zero-gradient issue inherent in top-1 MoEs.

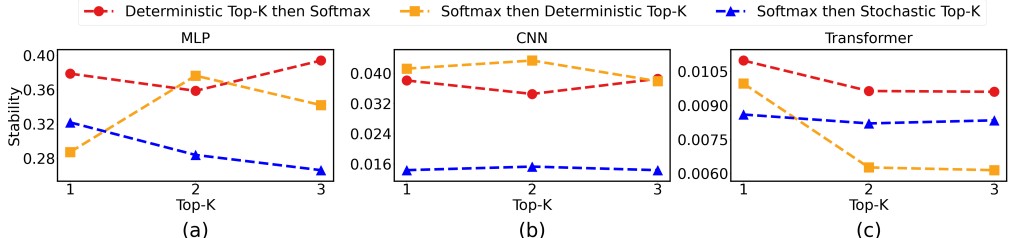

Figure 7: **Improved Stability in Soft Top-$K$ MoEs.** (a) MLP. (b) CNN. (c) Transformer. Each subplot presents the training stability $\epsilon_{\text{stab}}$ across different routing methods: original deterministic top-$K$ followed by softmax (red line), softmax followed by deterministic top-$K$ (orange line), and softmax followed by stochastic top-$K$ via sampling without replacement (blue line). The x-axis represents top-$K$, while the y-axis indicates the stability values. The stochastic top-$K$ routing (blue line) shows the highest stability, followed by the deterministic top-$K$ after softmax (orange line), and the least stable being the original deterministic top-$K$ routing (red line). This suggests that our soft top-$K$ routing provides the most stable training dynamics, especially in sparse configurations, significantly improving upon the traditional deterministic top-$K$ approach.

stochasticity into the expert selection process. These improvements are especially evident in highly sparse scenarios, where traditional top-$K$ approaches struggle with unstable training dynamics.

## 6 CONCLUSION

In this paper, we have explored the intricate balance between expert sparsity and training stability within MoEs. Through rigorous theoretical analysis, we have demonstrated that gradient smoothness acts as a pivotal factor in harmonizing these two aspects. Our research reveals that denser MoE configurations yield smoother gradients, thereby enhancing the stability of the training process. The empirical evidence presented across diverse architectures and datasets corroborates our theoretical insights, offering practical guidance for optimizing MoE architectures to achieve stable and efficient training. Additionally, we introduce a novel MoE structure that theoretically guarantees improved stability by addressing the inherent challenges posed by sparse expert selection. This work not only enriches the theoretical discourse on MoE stability but also provides actionable strategies for real-world MoE deployments.

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

## A   THE ILLUSTRATIVE EXPERIMENT

**Data**: We consider a straightforward scenario where the input data $x$ are uniformly distributed across the interval $[-1, 1]$, denoted as $U[-1, 1]$. This uniform distribution is selected to provide a controlled and interpretable setting for analyzing the behavior of the Mixture of Experts (MoE) model. The target output for all input data points is set to 0, simplifying the assessment of the model's optimization performance.

**Model**: The model employed in this experiment is a one-layer MoE consisting of two linear experts. The expert functions are defined as $f_1(\theta_1; x) = \theta_1 x$ and $f_2(\theta_2; x) = \theta_2 x$, where $\theta_1$ and $\theta_2$ are the respective parameters of the experts. The router function is fixed with equal probabilities $(p_1, p_2) = \left(\frac{1}{2}, \frac{1}{2}\right)$, ensuring that any input $x$ is equally likely to be routed to either $f_1$ or $f_2$. The output $y$ of the MoE varies depending on the top-$k$ setting. When top-$k = 2$, the output is calculated as a weighted sum of contributions from both experts: $y = p_1 f_1 + p_2 f_2$. For top-$k = 1$, the output is randomly selected as either $f_1$ or $f_2$. The loss function is defined as the squared 2-norm between the target value 0 and the output $y$, expressed as $l(\theta_1, \theta_2) = y^2$. The optimization is carried out using gradient descent with a fixed learning rate of 0.1, over 20 steps. This simplified MoE setup is designed to illustrate the relationship between expert sparsity, gradient smoothness, and training stability within a controlled experimental framework.

## B   PROOF OF THEOREM 2

In this sub-section, we present a detailed analysis leading to the proof of Theorem 1. Our approach begins by considering the structure of the MoE network and establishing bounds on the router probabilities $\mathcal{G}_{i,j}$. We derive the Jacobians of the MoE block output with respect to both the router and expert parameters, followed by computing the squared $l_2$-norms of these Jacobians. These computations allow us to determine the local Lipschitz constants of the whole network using the chain rule. Subsequently, we extend this analysis to the Hessians, which helps us establish the local smoothness constants. Finally, by combining these results with our assumptions and leveraging Theorem 3, we derive the bounds necessary to prove the stability result encapsulated in Theorem 1.

We begin by the following theorem introduced in (Hardt et al., 2016),

**Theorem 3** *(Upper bound on general stability).* *Assume that the loss function $\mathcal{L}(\cdot; \mathbf{x}) \in [0, 1]$ is convex, $\beta$-smooth, and $L$-Lipschitz for every $\mathbf{x}$. If we run mini-batch SGD with fixed step sizes $\alpha = 2/\beta$ for $T$ steps, then mini-batch SGD satisfies uniform stability with:*

$$\epsilon_{stab} \leq \frac{L^2 T}{\beta B}.$$

Theorem 3 reveals that the stability of mini-batch SGD improves as the Lipschitz constant $L$ decreases, which is consistent with the intuition that smoother loss landscapes contribute to more stable learning. This result is crucial for understanding how to control the trade-offs between learning rate, batch size, and smoothness in practical applications. Furthermore, it lays the groundwork for Theorems 1 and 2, which extend these stability considerations to the context of MoE models, specifically addressing the unique challenges posed by expert sparsity in these architectures.

For simplicity, we omit the tuples $(\mathbf{\Theta}; \mathbf{x})$, $(\mathbf{\Theta}_i^g; \mathbf{x})$, and $\left(\mathbf{\Theta}_{i,j}^f; \mathbf{x}\right)$. Consider a MoE network $\mathcal{F}$ composed of $N$ MoE blocks:

$$\mathcal{F} = \mathcal{F}_1 \circ \mathcal{F}_2 \circ \cdots \circ \mathcal{F}_N.$$

The MoE block function $\mathcal{F}_i$ is given by:

$$\mathcal{F}_i(\mathbf{\Theta}_i; \mathbf{x}) = \sum_{j \in \mathcal{T}_i} \mathcal{G}_{i,j}(\mathbf{\Theta}_i^g; \mathbf{x}) f_{i,j}\left(\mathbf{\Theta}_{i,j}^f; \mathbf{x}\right),$$

where the router probability function $\mathcal{G}_{i,j}$ is:

$$\mathcal{G}_{i,j}(\mathbf{\Theta}_i^g; \mathbf{x}) = \text{softmax}\left(\text{TopK}_i\left(g_{i,j}(\mathbf{\Theta}_i^g; \mathbf{x})\right)\right)$$

$$= \frac{\mathbf{1}_{j \in \mathcal{T}_i} \cdot \exp\left(g_{i,j}(\mathbf{\Theta}_i^g; \mathbf{x})\right)}{\sum_{k \in \mathcal{T}_i} \exp\left(g_{i,k}(\mathbf{\Theta}_i^g; \mathbf{x})\right)}.$$

Let $\mathcal{T}_i^{\text{full}}$ be the index set of all experts. We first show that $\mathcal{G}_{i,j} = O\left(\frac{1}{K_i}\right)$ if $j \in \mathcal{T}_i$. Starting with the upper bound:

$$
\begin{aligned}
\mathcal{G}_{i,j}\left(\boldsymbol{\Theta}_i^g; \mathbf{x}\right) &= \frac{\exp\left(g_{i,j}(\boldsymbol{\Theta}_i^g; \mathbf{x})\right)}{\sum_{k \in \mathcal{T}_i} \exp\left(g_{i,k}(\boldsymbol{\Theta}_i^g; \mathbf{x})\right)} \\
&\leq \frac{\max_{j \in \mathcal{T}_i^{\text{full}}} \exp\left(g_{i,j}(\boldsymbol{\Theta}_i^g; \mathbf{x})\right)}{K_i \min_{k \in \mathcal{T}_i^{\text{full}}} \exp\left(g_{i,k}(\boldsymbol{\Theta}_i^g; \mathbf{x})\right)} \\
&= O\left(\frac{1}{K_i}\right),
\end{aligned}
\tag{2}
$$

where the last equality follows because $g_{i,j}$ is bounded according to Assumption 1. Similarly, for the lower bound:

$$
\begin{aligned}
\mathcal{G}_{i,j}\left(\boldsymbol{\Theta}_i^g; \mathbf{x}\right) &= \frac{\exp\left(g_{i,j}(\boldsymbol{\Theta}_i^g; \mathbf{x})\right)}{\sum_{k \in \mathcal{T}_i} \exp\left(g_{i,k}(\boldsymbol{\Theta}_i^g; \mathbf{x})\right)} \\
&\geq \frac{\min_{j \in \mathcal{T}_i^{\text{full}}} \exp\left(g_{i,j}(\boldsymbol{\Theta}_i^g; \mathbf{x})\right)}{K_i \max_{k \in \mathcal{T}_i^{\text{full}}} \exp\left(g_{i,k}(\boldsymbol{\Theta}_i^g; \mathbf{x})\right)} \\
&= O\left(\frac{1}{K_i}\right).
\end{aligned}
\tag{3}
$$

Combining Equations (2) and (3), we establish that $\mathcal{G}_{i,j} = O\left(\frac{1}{K_i}\right)$ if $j \in \mathcal{T}_i$.

Next, we derive the Jacobian of the MoE block output $\mathcal{F}_i$ with respect to the router parameters $\boldsymbol{\Theta}_i^g$:

$$
\begin{aligned}
\frac{\partial \mathcal{F}_i}{\partial \boldsymbol{\Theta}_i^g} &= \sum_{j \in \mathcal{T}_i} \frac{\partial \mathcal{G}_{i,j}}{\partial \boldsymbol{\Theta}_i^g} f_{i,j} \\
&= \sum_{j \in \mathcal{T}_i} \left( \left( \left( \exp\left(g_{i,j}\right) \frac{\partial g_{i,j}}{\partial \boldsymbol{\Theta}_i^g} \left( \sum_{k \in \mathcal{T}_i} \exp\left(g_{i,k}\right) \right) - \right. \right. \right. \\
&\qquad \left. \left. \left. \exp\left(g_{i,j}\right) \left( \sum_{k \in \mathcal{T}_i} \exp\left(g_{i,k}\right) \frac{\partial g_{i,k}}{\partial \boldsymbol{\Theta}_i^g} \right) \right) \right/ \right. \\
&\qquad \left. \left( \sum_{k \in \mathcal{T}_i} \exp\left(g_{i,k}\right) \right)^2 f_{i,j} \right) \\
&= \sum_{j \in \mathcal{T}_i} \left( \left( \sum_{k \in \mathcal{T}} \left( \exp\left(g_{i,j} + g_{i,k}\right) \left( \frac{\partial g_{i,j}}{\partial \boldsymbol{\Theta}_i^g} - \frac{\partial g_{i,k}}{\partial \boldsymbol{\Theta}_i^g} \right) \right) \right) \right/ \\
&\qquad \left( \sum_{k \in \mathcal{T}_i} \exp\left(g_{i,k}\right) \right)^2 f_{i,j} \right) \\
&= \sum_{j \in \mathcal{T}_i} \sum_{k \in \mathcal{T}_i} \mathcal{G}_{i,j} \mathcal{G}_{i,k} \left( \frac{\partial g_{i,j}}{\partial \boldsymbol{\Theta}_i^g} - \frac{\partial g_{i,k}}{\partial \boldsymbol{\Theta}_i^g} \right) f_{i,j} \\
&= \sum_{j \in \mathcal{T}_i} \mathcal{G}_{i,j} \left( \sum_{k \in \mathcal{T}_i} \mathcal{G}_{i,k} \left( \frac{\partial g_{i,j}}{\partial \boldsymbol{\Theta}_i^g} - \frac{\partial g_{i,k}}{\partial \boldsymbol{\Theta}_i^g} \right) \right) f_{i,j}.
\end{aligned}
\tag{4}
$$

First, note that when $K_i = 1$, $\sum_{k \in \mathcal{T}_i} g_{i,k} \left( \frac{\partial g_{i,j}}{\partial \boldsymbol{\Theta}_i^g} - \frac{\partial g_{i,k}}{\partial \boldsymbol{\Theta}_i^g} \right) = g_{i,j} \left( \frac{\partial g_{i,j}}{\partial \boldsymbol{\Theta}_i^g} - \frac{\partial g_{i,j}}{\partial \boldsymbol{\Theta}_i^g} \right) = 0$, i.e., $\frac{\partial \mathcal{F}_i(\boldsymbol{\Theta}_i; \mathbf{x})}{\partial \boldsymbol{\Theta}_i^g} = 0$. This proves Proposition 1.

When $K_i > 1$, denote $\mathcal{G}_{i,j} \left( \sum_{k \in \mathcal{T}_i} \mathcal{G}_{i,k} \left( \frac{\partial g_{i,j}}{\partial \boldsymbol{\Theta}_i^g} - \frac{\partial g_{i,k}}{\partial \boldsymbol{\Theta}_i^g} \right) \right) f_{i,j}$ as $\mathcal{G}_{i,j} \mathbf{a}_j$ and $\mathcal{G}_{i,k} \left( \frac{\partial g_{i,j}}{\partial \boldsymbol{\Theta}_i^g} - \frac{\partial g_{i,k}}{\partial \boldsymbol{\Theta}_i^g} \right)$ as $\mathcal{G}_{i,k} \mathbf{b}_k$. To obtain the squared $l_2$-norm of $\frac{\partial \mathcal{F}_i}{\partial \boldsymbol{\Theta}_i^g}$, we first compute the squared $l_2$-norm of the inner

summation,

$$\left\| \sum_{k \in \mathcal{T}_i} \mathcal{G}_{i,k} \left( \frac{\partial g_{i,j}}{\partial \mathbf{\Theta}_i^g} - \frac{\partial g_{i,k}}{\partial \mathbf{\Theta}_i^g} \right) \right\|_2^2 = \sum_{k \in \mathcal{T}_i} \mathcal{G}_{i,k}^2 \|\mathbf{b}_k\|_2^2 +$$

$$2 \sum_{p,q \in \mathcal{T}_i} \mathcal{G}_{i,p} \mathcal{G}_{i,q} \mathbf{b}_p^\top \mathbf{b}_q$$

$$= \sum_{k \in \mathcal{T}_i} O\left( \frac{1}{K_i^2} \right) \|\mathbf{b}_k\|_2^2 +$$

$$2 \sum_{p,q \in \mathcal{T}_i} O\left( \frac{1}{K_i^2} \right) \mathbf{b}_p^\top \mathbf{b}_q$$

$$= O\left( \frac{1}{K_i} \right).$$

And hence we can compute the squared $l_2$-norm of the outer summation,

$$\left\| \frac{\partial \mathcal{F}_i}{\partial \mathbf{\Theta}_i^g} \right\|_2^2 = \left\| \sum_{j \in \mathcal{T}_i} \mathcal{G}_{i,j} \left( \sum_{k \in \mathcal{T}_i} \mathcal{G}_{i,k} \left( \frac{\partial g_{i,j}}{\partial \mathbf{\Theta}_i^g} - \frac{\partial g_{i,k}}{\partial \mathbf{\Theta}_i^g} \right) \right) f_{i,j} \right\|_2^2$$

$$= \sum_{j \in \mathcal{T}_i} \mathcal{G}_{i,k}^2 \|\mathbf{a}_j\|_2^2 + 2 \sum_{p,q \in \mathcal{T}_i} \mathcal{G}_{i,p} \mathcal{G}_{i,q} \mathbf{a}_p^\top \mathbf{a}_q$$

$$= \sum_{j \in \mathcal{T}_i} \mathcal{G}_{i,k}^2 O\left( \frac{1}{K_i} \right) \|f_{i,j}\|_2^2 +$$

$$2 \sum_{p,q \in \mathcal{T}_i} \mathcal{G}_{i,p} \mathcal{G}_{i,q} O\left( \frac{1}{K_i} \right) f_{i,p}^\top f_{i,q} \tag{5}$$

$$= \sum_{k \in \mathcal{T}_i} O\left( \frac{1}{K_i^3} \right) \|f_{i,j}\|_2^2 +$$

$$2 \sum_{p,q \in \mathcal{T}_i} O\left( \frac{1}{K_i^3} \right) f_{i,p}^\top f_{i,q}$$

$$= O\left( \frac{1}{K_i^2} \right).$$

The Jacobian with respect to the expert parameters $\mathbf{\Theta}_{i,j}^f$ is given by:

$$\frac{\partial \mathcal{F}_i}{\partial \mathbf{\Theta}_{i,j}^f} = \begin{cases} \mathcal{G}_{i,j} \frac{\partial f_{i,j}}{\partial \mathbf{\Theta}_{i,j}^f}, & \text{if } j \in \mathcal{T}_i, \\ 0, & \text{otherwise.} \end{cases} \tag{6}$$

For $j \in \mathcal{T}_i$, the squared $l_2$-norm of $\frac{\partial \mathcal{F}_i}{\partial \mathbf{\Theta}_{i,j}^f}$ is as follows:

$$\left\| \frac{\partial \mathcal{F}_i}{\partial \mathbf{\Theta}_{i,j}^f} \right\|_2^2 = \mathcal{G}_{i,j}^2 \left\| \frac{\partial f_{i,j}}{\partial \mathbf{\Theta}_{i,j}^f} \right\|_2^2 = O\left( \frac{1}{K_i^2} \right). \tag{7}$$

The Jacobian with respect to the input $\mathbf{z}_i$ is derived similarly:

$$\frac{\partial \mathcal{F}_i}{\partial \mathbf{z}_i} = \sum_{j \in \mathcal{T}_i} \left( \mathcal{G}_{i,j} \left( \sum_{k \in \mathcal{T}_i} \mathcal{G}_{i,k} \left( \frac{\partial g_{i,j}}{\partial \mathbf{z}_i} - \frac{\partial g_{i,k}}{\partial \mathbf{z}_i} \right) \right) f_{i,j} +$$

$$\mathcal{G}_{i,j} \frac{\partial f_{i,j}}{\partial \mathbf{z}_i} \right). \tag{8}$$

Noticing that $\|\mathcal{G}_{i,j}\left(\sum_{k\in\mathcal{T}_i}\mathcal{G}_{i,k}\left(\frac{\partial g_{i,j}}{\mathbf{z}_i}-\frac{\partial g_{i,k}}{\partial\mathbf{z}_i}\right)\right)f_{i,j}\|_2^2 = O\left(\frac{1}{K_i^3}\right)$ and $\left\|\mathcal{G}_{i,j}\frac{\partial f_{i,j}}{\partial\mathbf{z}_i}\right\|_2^2 = O\left(\frac{1}{K_i^2}\right)$.

Denote $\mathcal{G}_{i,j}\left(\sum_{k\in\mathcal{T}_i}\mathcal{G}_{i,k}\left(\frac{\partial g_{i,j}}{\mathbf{z}_{i-1}}-\frac{\partial g_{i,k}}{\partial\mathbf{z}_{i-1}}\right)\right)f_{i,j}+\mathcal{G}_{i,j}\frac{\partial f_{i,j}}{\partial\mathbf{z}_i}$ as $\mathbf{c}_j$, we can compute $\|\mathbf{c}_j\|_2^2 = O\left(\frac{1}{K_i^3}\right)+O\left(\frac{1}{K_i^2}\right)+2O\left(\frac{1}{K_i^{2.5}}\right) = O\left(\frac{1}{K_i^2}\right)$ and

$$
\begin{aligned}
\left\|\frac{\partial\mathcal{F}_i}{\partial\mathbf{z}_i}\right\|_2^2 &= \left\|\sum_{j\in\mathcal{T}_i}\mathbf{c}_j\right\|_2^2 \\
&= \sum_{j\in\mathcal{T}_i}\|\mathbf{c}_j\|_2^2 + 2\sum_{p,q\in\mathcal{T}_i}\mathbf{c}_p^\top\mathbf{c}_q \\
&= \sum_{j\in\mathcal{T}_i}O\left(\frac{1}{K_i^2}\right) + 2\sum_{p,q\in\mathcal{T}_i}O\left(\frac{1}{K_i^2}\right) \\
&= O\left(\frac{1}{K_i}\right).
\end{aligned}
\tag{9}
$$

Using the chain rule $\frac{\partial\mathcal{L}}{\partial\mathbf{\Theta}_i} = \frac{\partial\mathcal{L}}{\partial\mathbf{z}_N}\left(\prod_{j=i+1}^N\frac{\partial\mathcal{F}_j}{\partial\mathbf{z}_j}\right)\frac{\partial\mathcal{F}_i}{\partial\mathbf{\Theta}_i}$ and Equations (5), (6), and (9), we can obtain the squared $l_2$-norms of the Jacobians of the entire MoE network:

$$
\begin{aligned}
\left\|\frac{\partial\mathcal{L}}{\partial\mathbf{\Theta}_i^g}\right\|_2^2 &= \left\|\frac{\partial\mathcal{L}}{\partial\mathbf{z}_N}\right\|_2^2\left(\prod_{j=i+1}^N\left\|\frac{\partial\mathcal{F}_j}{\partial\mathbf{z}_j}\right\|_2^2\right)\left\|\frac{\partial\mathcal{F}_i}{\partial\mathbf{\Theta}_i^g}\right\|_2^2 \\
&= O(1)\left(\prod_{j=i+1}^N O\left(\frac{1}{K_j}\right)\right)O\left(\frac{1}{K_i^2}\right) \\
&= O\left(\frac{1}{K_i^2\prod_{j=i+1}^N K_j}\right),
\end{aligned}
\tag{10}
$$

and

$$
\begin{aligned}
\left\|\frac{\partial\mathcal{L}}{\partial\mathbf{\Theta}_{i,j}^f}\right\|_2^2 &= \left\|\frac{\partial\mathcal{L}}{\partial\mathbf{z}_N}\right\|_2^2\left(\prod_{j=i+1}^N\left\|\frac{\partial\mathcal{F}_j}{\partial\mathbf{z}_j}\right\|_2^2\right)\left\|\frac{\partial\mathcal{F}_i}{\partial\mathbf{\Theta}_{i,j}^f}\right\|_2^2 \\
&= O(1)\left(\prod_{j=i+1}^N O\left(\frac{1}{K_j}\right)\right)O\left(\frac{1}{K_i^2}\right) \\
&= O\left(\frac{1}{K_i^2\prod_{j=i+1}^N K_j}\right).
\end{aligned}
\tag{11}
$$

Since $\frac{\partial \mathcal{F}_i}{\partial \Theta_i} := \left( \frac{\partial \mathcal{F}_i}{\partial \Theta_i^g}, \frac{\partial \mathcal{F}_i}{\partial \Theta_{i,1}^f}, \ldots, \frac{\partial \mathcal{F}_i}{\partial \Theta_{i,N_i}^f} \right)$ and $\frac{\partial \mathcal{L}}{\partial \Theta} := \left( \frac{\partial \mathcal{L}}{\partial \Theta_1}, \frac{\partial \mathcal{L}}{\partial \nabla_{\Theta_2}}, \ldots, \frac{\partial \mathcal{L}}{\partial \Theta_N} \right)$, we have

$$
\begin{aligned}
\left\| \frac{\partial \mathcal{L}}{\partial \Theta} \right\|_2^2 &= \sum_{i=1}^N \left\| \frac{\partial \mathcal{L}}{\partial \Theta_i} \right\|_2^2 \\
&= \sum_{i=1}^N \left( \left\| \frac{\partial \mathcal{F}_i}{\partial \Theta_i^g} \right\|_2^2 + \sum_{j \in \mathcal{T}_i} \left\| \frac{\partial \mathcal{F}_i}{\partial \Theta_{i,j}^f} \right\|_2^2 \right) \\
&= \sum_{i=1}^N \left( O \left( \frac{1}{K_i^2 \prod_{j=i+1}^N K_j} \right) + \right. \\
&\qquad \left. \sum_{j \in \mathcal{T}_i} O \left( \frac{1}{K_i^2 \prod_{j=i+1}^N K_j} \right) \right) \\
&= \sum_{i=1}^N \left( O \left( \frac{1}{K_i^2 \prod_{j=i+1}^N K_j} \right) + \right. \\
&\qquad \left. O \left( \frac{1}{\prod_{j=i+1}^N K_j} \right) \right) \\
&= \sum_{i=1}^N \left( O \left( \frac{1}{\prod_{j=i}^N K_j} \right) \right) \\
&= O \left( \frac{1}{K_N} \right)
\end{aligned}
\tag{12}
$$

Therefore, $\left\| \frac{\partial \mathcal{L}}{\partial \Theta} \right\|_2 = O \left( \frac{1}{\sqrt{K_N}} \right)$ and the local Lipschitzness constant $L = \max_{\Theta \in \mathcal{B}_\epsilon(\tilde{\Theta})} \left\| \frac{\partial \mathcal{L}}{\partial \Theta} \right\|_2 = O \left( \frac{1}{\sqrt{K_N}} \right)$.

Following the same approach used to prove Lemma 1 but computing Hessians instead, we first derive the Hessians of the router and experts in each MoE block:

$$
\begin{aligned}
\frac{\partial^2 \mathcal{F}_i}{\left( \partial \Theta_i^g \right)^2} &= \frac{\partial \left( \frac{\partial \mathcal{F}_i}{\partial \Theta_i^g} \right)}{\partial \Theta_i^g} \\
&= \sum_{j \in \mathcal{T}_i} \mathcal{G}_{i,j} f_{i,j} \left( \left( \sum_{k \in \mathcal{T}_i} \mathcal{G}_{i,k} \left( \frac{\partial g_{i,j}}{\partial \Theta_i^g} - \frac{\partial g_{i,k}}{\partial \Theta_i^g} \right) \right)^2 + \right. \\
&\qquad \sum_{k \in \mathcal{T}_i} \left( \mathcal{G}_{i,k}^2 \left( \frac{\partial g_{i,j}}{\partial \Theta_i^g} - \frac{\partial g_{i,k}}{\partial \Theta_i^g} \right)^2 + \right. \\
&\qquad \left. \left. \mathcal{G}_{i,k} \left( \frac{\partial^2 g_{i,j}}{\left( \partial \Theta_i^g \right)^2} - \frac{\partial^2 g_{i,k}}{\left( \partial \Theta_i^g \right)^2} \right) \right) \right),
\end{aligned}
\tag{13}
$$

and

$$
\frac{\partial^2 \mathcal{F}_i}{\left( \partial \Theta_{i,j}^f \right)^2} = \begin{cases} \mathcal{G}_{i,j} \frac{\partial^2 f_{i,j}}{\left( \partial \Theta_{i,j}^f \right)^2}, & \text{if } j \in \mathcal{T}_i, \\ 0, & \text{otherwise.} \end{cases}
\tag{14}
$$

Computing the squared $l_2$-norms of the above two Hessians:

$$
\begin{aligned}
\left\| \frac{\partial^2 \mathcal{F}_i}{(\partial \boldsymbol{\Theta}_i^g)^2} \right\|_2^2 &= \sum_{j \in \mathcal{T}_i} O\left(\frac{1}{K_i}\right) \left( \left( \sum_{k \in \mathcal{T}_i} O\left(\frac{1}{K_i}\right) \right)^2 + \right. \\
&\qquad \left. \sum_{k \in \mathcal{T}_i} \left( O\left(\frac{1}{K_i^2}\right) + O\left(\frac{1}{K_i}\right) \right) \right), \\
&= \sum_{j \in \mathcal{T}_i} O\left(\frac{1}{K_i^2}\right) \left( O\left(\frac{1}{K_i^2}\right) + O\left(\frac{1}{K_i}\right) \right) \\
&= \sum_{j \in \mathcal{T}_i} O\left(\frac{1}{K_i^3}\right) \\
&= O\left(\frac{1}{K_i^2}\right),
\end{aligned}
\tag{15}
$$

and

$$
\left\| \frac{\partial^2 \mathcal{F}_i}{\left(\partial \boldsymbol{\Theta}_{i,j}^f\right)^2} \right\|_2^2 = \mathcal{G}_{i,j}^2 \left\| \frac{\partial^2 f_{i,j}}{\left(\partial \boldsymbol{\Theta}_{i,j}^f\right)^2} \right\|_2^2 = O\left(\frac{1}{K_i^2}\right).
\tag{16}
$$

Again, using the chain rule and Equations (9), (15), and (16), we obtain the $l_2$-norms of the Hessians of the whole MoE network:

$$
\begin{aligned}
\left\| \frac{\partial^2 \mathcal{L}}{(\partial \boldsymbol{\Theta}_i^g)^2} \right\|_2^2 &= \left\| \frac{\partial \mathcal{L}}{\partial \mathbf{z}_N} \right\|_2^2 \left( \prod_{j=i+1}^{N} \left\| \frac{\partial \mathcal{F}_j}{\partial \mathbf{z}_j} \right\|_2^2 \right) \left\| \frac{\partial^2 \mathcal{F}_i}{(\partial \boldsymbol{\Theta}_i^g)^2} \right\|_2^2 \\
&= O(1) \left( \prod_{j=i+1}^{N} O\left(\frac{1}{K_j}\right) \right) O\left(\frac{1}{K_i^2}\right) \\
&= O\left( \frac{1}{K_i^2 \prod_{j=i+1}^{N} K_j} \right),
\end{aligned}
\tag{17}
$$

and

$$
\begin{aligned}
\left\| \frac{\partial^2 \mathcal{L}}{\left(\partial \boldsymbol{\Theta}_{i,j}^f\right)^2} \right\|_2^2 &= \left\| \frac{\partial \mathcal{L}}{\partial \mathbf{z}_N} \right\|_2^2 \left( \prod_{j=i+1}^{N} \left\| \frac{\partial \mathcal{F}_j}{\partial \mathbf{z}_j} \right\|_2^2 \right) \left\| \frac{\partial^2 \mathcal{F}_i}{\left(\partial \boldsymbol{\Theta}_{i,j}^f\right)^2} \right\|_2^2 \\
&= O(1) \left( \prod_{j=i+1}^{N} O\left(\frac{1}{K_j}\right) \right) O\left(\frac{1}{K_i^2}\right) \\
&= O\left( \frac{1}{K_i^2 \prod_{j=i+1}^{N} K_j} \right),
\end{aligned}
\tag{18}
$$

Following the same process used to derive Equation (12) and noting that $\frac{\partial^2 \mathcal{F}_i}{(\partial \boldsymbol{\Theta}_i)^2} :=$ $\left( \frac{\partial^2 \mathcal{F}_i}{(\partial \boldsymbol{\Theta}_i^g)^2}, \frac{\partial^2 \mathcal{F}_i}{(\partial \boldsymbol{\Theta}_{i,j}^f)^2}, \ldots, \frac{\partial^2 \mathcal{F}_i}{(\partial \boldsymbol{\Theta}_{i,j}^f)^2} \right)$ and $\frac{\partial^2 \mathcal{L}}{(\partial \boldsymbol{\Theta})^2} := \left( \frac{\partial^2 \mathcal{L}}{(\partial \boldsymbol{\Theta}_1)^2}, \frac{\partial^2 \mathcal{L}}{(\partial \boldsymbol{\Theta}_2)^2}, \ldots, \frac{\partial^2 \mathcal{L}}{(\partial \boldsymbol{\Theta}_N)^2} \right)$, we obtain:

$$
\begin{aligned}
\left\| \frac{\partial^2 \mathcal{L}}{(\partial \boldsymbol{\Theta})^2} \right\|_2^2 &= \sum_{i=1}^{N} \left\| \frac{\partial^2 \mathcal{L}}{(\partial \boldsymbol{\Theta}_i)^2} \right\|_2^2 \\
&= \sum_{i=1}^{N} \left( \left\| \frac{\partial^2 \mathcal{F}_i}{(\partial \boldsymbol{\Theta}_i^g)^2} \right\|_2^2 + \sum_{j \in \mathcal{T}_i} \left\| \frac{\partial^2 \mathcal{F}_i}{(\partial \boldsymbol{\Theta}_{i,j}^f)^2} \right\|_2^2 \right) \\
&= \sum_{i=1}^{N} \left( O\left( \frac{1}{K_i^2 \prod_{j=i+1}^{N} K_j} \right) + \right. \\
&\qquad \left. \sum_{j \in \mathcal{T}_i} O\left( \frac{1}{K_i^2 \prod_{j=i+1}^{N} K_j} \right) \right) \\
&= \sum_{i=1}^{N} \left( O\left( \frac{1}{K_i^2 \prod_{j=1+1}^{N} K_j} \right) + \right. \\
&\qquad \left. O\left( \frac{1}{\prod_{j=i+1}^{N} K_j} \right) \right) \\
&= \sum_{i=1}^{N} \left( O\left( \frac{1}{\prod_{j=i}^{N} K_j} \right) \right) \\
&= O\left( \frac{1}{K_N} \right)
\end{aligned}
\tag{19}
$$

Therefore, $\left\| \frac{\partial^2 \mathcal{L}}{(\partial \boldsymbol{\Theta})^2} \right\|_2 = O\left( \frac{1}{\sqrt{K_N}} \right)$ and the local smoothness constant $\beta = \max_{\boldsymbol{\Theta} \in \mathcal{B}_\epsilon(\tilde{\boldsymbol{\Theta}})} \left\| \frac{\partial^2 \mathcal{L}}{(\partial \boldsymbol{\Theta})^2} \right\|_2 = O\left( \frac{1}{\sqrt{K_N}} \right)$. This completes the proof of Lemma 1.

Finally, using Theorem 1, Lemma 1, together with Assumption 1),we have:

$$
\epsilon_{\text{stab}}^{\text{MoE}} \leq \frac{\left( L^{\text{MoE}} \right)^2 T}{\beta^{\text{MoE}} B} = O\left( \frac{T}{\sqrt{K_N} B} \right),
$$

which completes the proof of Theorem 2.

## C  PROOF OF THEOREM 3

The proof of Theorem 3 closely follows the structure of the proof of Theorem 1, with the primary differences lying in: (1) the big $O$ bound of the modified router probability function $\tilde{\mathcal{G}}_{i,j}$, and (2) the derivation of the Jacobians $\frac{\partial \mathcal{L}_i}{\partial \boldsymbol{\Theta}_{i,j}^g}$, $\frac{\partial \tilde{\mathcal{F}}_i}{\partial \boldsymbol{\Theta}_{i,j}^g}$, $\frac{\partial \tilde{\mathcal{F}}_i}{\partial \mathbf{z}_i}$, and the Hessians $\frac{\partial^2 \mathcal{L}_i}{(\partial \boldsymbol{\Theta}_{i,j}^g)^2}$, $\frac{\partial^2 \tilde{\mathcal{F}}_i}{(\partial \boldsymbol{\Theta}_{i,j}^g)^2}$ with respect to the independent router parameters $\boldsymbol{\Theta}_{i,j}^g$.

Recall that the modified MoE block function $\tilde{\mathcal{F}}_i$ is given by:

$$
\tilde{\mathcal{F}}_i \left( \boldsymbol{\Theta}_i; \mathbf{x} \right) = \sum_{j \in \tilde{\mathcal{T}}_i} \tilde{\mathcal{G}}_{i,j} \left( \boldsymbol{\Theta}_i^g; \mathbf{x} \right) f_{i,j} \left( \boldsymbol{\Theta}_{i,j}^f; \mathbf{x} \right),
$$

where $\tilde{T}_i$ is the set of expert indices selected by soft top-$K$ and the modified router probability function $\tilde{\mathcal{G}}_{i,j}$ is:

$$\tilde{\mathcal{G}}_{i,j}\left(\mathbf{\Theta}_{i,j}^g; \mathbf{x}\right) = \text{SoftTopK}_i\left(\text{softmax}\left(\tilde{g}_{i,j}(\mathbf{\Theta}_{i,j}^g; \mathbf{x})\right)\right)$$

$$= \mathbf{1}_{j \in \tilde{\mathcal{T}}_i} \cdot \frac{\exp\left(\tilde{g}_{i,j}(\mathbf{\Theta}_{i,j}^g; \mathbf{x})\right)}{\sum_{k=1}^{K_i^{\text{full}}} \exp\left(\tilde{g}_{i,k}(\mathbf{\Theta}_{i,j}^g; \mathbf{x})\right)}.$$

We begin by providing a new big $O$ bound on $\tilde{\mathcal{G}}_{i,j}$. Using the same reasoning as in the derivation of Equations (2) and (3), replacing $K_i$ with $K_i^{\text{full}}$, we obtain $\tilde{\mathcal{G}}_{i,j} = O\left(\frac{1}{K_i^{\text{full}}}\right)$.

Next, we derive the Jacobian of the router probability function $\tilde{\mathcal{G}}_{i,j}$ with respect to the router parameters $\mathbf{\Theta}_{i,k}^g$ (where $j$ might be different from $k$):

$$\begin{aligned}
\frac{\partial \tilde{\mathcal{F}}_i}{\partial \mathbf{\Theta}_{i,k}^g} &= \sum_{j \in \tilde{\mathcal{T}}_i} \frac{\partial \tilde{\mathcal{G}}_{i,j}}{\partial \mathbf{\Theta}_{i,k}^g} f_{i,j} \\
&= \sum_{j \in \tilde{\mathcal{T}}_i} \left( \mathbf{1}_{j=k} \cdot \tilde{\mathcal{G}}_{i,j}(1 - \tilde{\mathcal{G}}_{i,j}) \frac{\partial \tilde{\mathcal{G}}_{i,j}}{\partial \mathbf{\Theta}_{i,k}^g} f_{i,j} - \right. \\
&\qquad\qquad \left. \mathbf{1}_{j \neq k} \cdot \tilde{\mathcal{G}}_{i,j} \tilde{\mathcal{G}}_{i,k} \frac{\partial \tilde{\mathcal{G}}_{i,j}}{\partial \mathbf{\Theta}_{i,k}^g} f_{i,j} \right) \\
&= \sum_{j \in \tilde{\mathcal{T}}_i} \tilde{\mathcal{G}}_{i,j} \left( \delta_{j,k} - \tilde{\mathcal{G}}_{i,k} \right) f_{i,j} \frac{\partial \tilde{\mathcal{G}}_{i,j}}{\partial \mathbf{\Theta}_{i,k}^g},
\end{aligned} \tag{20}$$

where $\delta_{j,k} = 1$ if $j = k$, and otherwise $\delta_{j,k} = 0$.

By comparing Equation (4) with (20), we observe that the term $\sum_{k \in \mathcal{T}_i} \mathcal{G}_{i,k} \left( \frac{\partial g_{i,j}}{\partial \mathbf{\Theta}_i^g} - \frac{\partial g_{i,k}}{\partial \mathbf{\Theta}_i^g} \right)$, which is responsible for zero router gradients when top-$K = 1$, is absent in Equation (20). This indicates that the modified MoE is immune to the zero-gradient issue, regardless of the value of top-$K$.

Furthermore, the Hessian of the router probability function $\tilde{\mathcal{G}}_{i,j}$ with respect to the router parameters $\mathbf{\Theta}_{i,k}^g$ (where $j$ might be different from $k$) is derived as follows:

$$\begin{aligned}
\frac{\partial^2 \tilde{\mathcal{F}}_i}{\left(\partial \mathbf{\Theta}_{i,k}^g\right)^2} &= \frac{\partial \left( \frac{\partial \tilde{\mathcal{F}}_i}{\partial \mathbf{\Theta}_{i,k}^g} \right)}{\partial \mathbf{\Theta}_{i,k}^g} \\
&= \sum_{j \in \tilde{\mathcal{T}}_i} f_{i,j} \left( \left( \tilde{\mathcal{G}}_{i,j} \left( \delta_{j,k} - \tilde{\mathcal{G}}_{i,k} \right) \frac{\partial \tilde{\mathcal{G}}_{i,j}}{\partial \mathbf{\Theta}_{i,k}^g} \right)^2 + \right. \\
&\qquad \left( \delta_{j,k} \tilde{\mathcal{G}}_{i,j} - \delta_{j,k} \tilde{\mathcal{G}}_{i,j}^2 + \tilde{\mathcal{G}}_{i,k}^3 \frac{\partial \tilde{\mathcal{G}}_{i,j}}{\partial \mathbf{\Theta}_{i,k}^g} \right) \frac{\partial \tilde{\mathcal{G}}_{i,j}}{\partial \mathbf{\Theta}_{i,k}^g} + \\
&\qquad \left. \tilde{\mathcal{G}}_{i,j} \left( \delta_{j,k} - \tilde{\mathcal{G}}_{i,k} \right) \frac{\partial^2 \tilde{\mathcal{G}}_{i,j}}{\left(\partial \mathbf{\Theta}_{i,k}^g\right)^2} \right).
\end{aligned} \tag{21}$$

Using the same reasoning for deriving the $l_2$-norms of the Jacobians and Hessians of the entire network, as in Equations (12) and (19), we obtain

$$
\left\| \frac{\partial \mathcal{L}}{\partial \mathbf{\Theta}} \right\|_2 = O \left( \frac{1}{\sqrt{K_N^{\text{full}}}} \right),
$$

$$
\left\| \frac{\partial^2 \mathcal{L}}{(\partial \mathbf{\Theta})^2} \right\|_2^2 = O \left( \frac{1}{\sqrt{K_N^{\text{full}}}} \right).
$$

(22)

Finally, applying Theorem 2, Lemma 1, together with Assumption 1, we derive:

$$
\epsilon_{\text{stab}}^{\text{mod-MoE}} \leq \frac{\left( L^{\text{mod-MoE}} \right)^2 T}{\beta^{\text{mod-MoE}} B} = O \left( \frac{T}{\sqrt{K_N^{\text{full}}} B} \right),
$$

which completes the proof of Theorem 3.

