# OpenReview forum: "Smoothness Bridges Sparsity and Stability in MoEs"
_ICLR.cc/2025/Conference — Submitted to ICLR 2025_

### Official Review · Reviewer_a5Fm · 2024-10-20

**Soundness:** 2
**Presentation:** 2
**Contribution:** 3
**Rating:** 3
**Confidence:** 4

**Summary:**

This paper investigates the sparsity-stability trade-off in Mixture-of-Experts (MoE) by providing a theoretical framework linking expert sparsity to training stability via gradient smoothness. The authors derive an upper bound on training stability, demonstrating that activating more experts improves stability at the cost of reduced sparsity. They introduce another MoE architecture with multi-headed routing and Gumbel-softmax sampling. The theoretical claims are validated through experiments across various architectures (MLPs, CNNs, Transformers) and datasets.

**Strengths:**

1. *Significance*: The focus on the sparsity-stability trade-off in MoE is an important and relatively under-explored problem. The proposed stability metric is insightful and has potential value to the community.
2. *Insightful Observations*: The experiments, which include observations of variance in loss, smoothness, and stability across various architectures, offer valuable insights and align well with the theoretical claims.

**Weaknesses:**

1. *Questionable Theoretical Assumptions*: Sparsely activated MoE with Top-K routers is known to be discontinuous, yet the authors assume Lipschitz continuity as the theoretical foundation for the entire framework. This assumption, along with the reliance on results from continuous optimization, is a *major concern* and weakens the theoretical basis of the paper.
2. *Clarity in Methodology*: The description of the methodology, specifically in Section 4.2, is overly concise. The multi-headed routing is not clearly explained—what distinguishes it from concatenating the weights into a single head?
3. *Novelty in the Proposed Method*: The Gumbel-softmax sampling technique is well-established and already implemented in mainstream MoE repositories, such as [Microsoft Deepspeed](https://github.com/microsoft/DeepSpeed/blob/master/deepspeed/moe/sharded_moe.py) , raising concerns about the novelty of the method.
4. *Small Experiment Scale*: The experiments are limited to small models and datasets, making it unclear whether the claims hold for large-scale models with billions of parameters, which are the primary applications of MoEs.

**Questions:**

1. In Figure 1, the initial point seems to be $(\theta_1,\theta_2)=(-1.5,1.5)$, where the output should be $y = 0.5f_1 + 0.5f_2 = 0.5(\theta_1x + \theta_2x) \equiv 0$, and the loss $l = 0$ for top-$k = 2$ according to Appendix A. Can the authors explain some critical points in this graph more clearly?
2. In Figure 6, the gradient of router weights in "Single-Headed" should be zero when $K=1$ according to Proposition 1. Why do the MLP and CNN exhibit nonzero gradients? Does the multi-headed MoE in this figure use soft Top-K or deterministic Top-K?
3. The theoretical results are derived for "Deterministic Top-K then Softmax." What would the theoretical outcome be for the alternative approach of "Softmax then Deterministic Top-K"?
4. How does the proposed method compare in terms of final loss or downstream performance against the baselines?
5. Typos: Appendix B should be the proof for Theorem 1, and Appendix C should correspond to Theorem 2.

---

### Official Review · Reviewer_ZAxu · 2024-10-26

**Soundness:** 2
**Presentation:** 3
**Contribution:** 2
**Rating:** 3
**Confidence:** 4

**Summary:**

In this paper, the authors explore the underlying relationship between expert sparsity and training stability of the Mixture-of-Experts (MoE) model. In particular, they demonstrate theoretically and empirically that activating more experts improves the gradient smoothness and the training stability but reducing the expert sparsity and vice versa. Then, they propose a new routing mechanism, which incorporates independent router heads and a soft top-K expert avtivation via sampling without replacement, in the MoE to smooth the gradient landscape without hurting the expert sparsity.

**Strengths:**

1. The relationship between the expert sparsity and the training stability of the MoE is of interest.

2. The presentation of this paper is good, which makes the paper easy to follow.

**Weaknesses:**

1. Potential mathematical errors: I have a major concern about the index set $\mathcal{T}_i$ defined in lines 152-156. Based on that definition and the original one in [1], I think the index set $\mathcal{T}_i$ should depend on the input values and the router values (see equation (5) in [1]). Moreover, due to the structure of the TopK function, the MoE output $\mathcal{F}_i$ is non-differentiable with respect to the router parameters as stating in [1]. However, the proof of Theorem 1 involves the Jacobian of the MoE output $\mathcal{F}_i$ with respect to the router parameters, which really confuses me. Please correct me if I am wrong.

I suggest that the authors should provide an explicit mathematical formulation for the index set $\mathcal{T}_i$ rather than just defining it in words, and then address my above concern. I think the results in the paper would be very helpful and I would consider raising my rating if this concern is addressed.

2. The authors should briefly introduce the Gumbel-softmax sampling and include some references for readers who do not have background on it to understand.

3. Typos:  The titles of Appendices B and C should be Proof of Theorem 1 and Proof of Theorem 2, respectively.

**References**

[1] Noam Shazeer, Azalia Mirhoseini, Krzysztof Maziarz, Andy Davis, Quoc Le, Geoffrey Hinton, and Jeff Dean. Outrageously large neural networks: The sparsely-gated mixture-of-experts layer. In ICLR, 2017.

**Questions:**

1. In the proposed gating $\tilde{\mathcal{G}}_{i,j}$, I see that for an MoE block $i$, the gating for the experts does not share parameters. Therefore, I am concerned that the sum of gating values in the MoE block might not sum up to one. Does this cause any challenges? Please correct me if I am wrong.

---

### Official Review · Reviewer_nCZP · 2024-11-02

**Soundness:** 3
**Presentation:** 2
**Contribution:** 3
**Rating:** 5
**Confidence:** 4

**Summary:**

This is a novel work on theoretically explaining the stability issues in MoE models. This work models training stability and uses gradient smoothness to establish a quantitative relationship between stability and expert sparsity. They also propose a solution to solve the trade-off that arises with increased sparsity and stability via stochastic top-k via Gumbel-softmax sampling and also suggest multi-headed routing to counter the zero gradients issue in top-1 MoEs. Their theoretical analysis is supported by related experimental evaluation.

**Strengths:**

1. Bridging expert sparsity and training stability with gradient smoothness is quite a novel idea. The authors are also able to theoretically justify the connection and model the training stability . In general, a mathematical approach to studying stability issues in MoEs is novel.
2. They also analyze analytically the issue of zero router gradients for top-1 MoEs which is a very prevalent problem and suggest a solution that seems feasible and is theoretically well motivated.
3. Smoothing gradient landscape with soft top-K selection via sampling without replacement is an interesting proposed solution and has a theoretical justification that hints about the trade-off between sparsity & stability. They mathematically model this relationship as well (Lemma 1)
4. Their theoretical analysis is well reflected by experimental evaluation of the claims.

**Weaknesses:**

Major weakness:
1. I find that there could have been more focus on describing your proposed approach, especially about stochastic top-k via Gumbel-softmax sampling. There is no mention of why this method is preferred over other sampling methods or why this improves over other methods like differentiable top-k selection for soft routing. Additionally, I am not confident how computationally efficient this is. (I also don't see any mathematical definition or description of your approach of stochastic top-k via Gumbel-softmax sampling)
2. Experiments:
- A very small learning rate is used with transformer models, which is not very practical. Does the analysis still stand with a practical learning rate?
- Large-scale evaluation is lacking. Cannot comment on how tractable the suggested approach is. In general, I would have liked to see how pronounced the issue of stability is for large language model families whose behavior is often studied in MoE research.
- Quantitative analysis of how improving stability leads to improvement in model performance/loss values etc. It is not clear currently if the solution proposed has any practicality.
3. Also, I wonder why the authors do not mention or compare against router z-loss which is known to improve over instability issues (or other existing router design-based stability approaches). Even if there are ideological differences I think it should be made clear.
4. Insufficient/not robust related sections. It could be improved and have more coverage.
5. Multi-Headed MoEs are claimed to eliminate zero router gradients - I am not sure if this is a novel idea. How is it different from Xun et. al [2024]?

References:
1. Xun Wu, Shaohan Huang, Wenhui Wang, Furu Wei, "Multi-Head Mixture-of-Experts", https://arxiv.org/abs/2404.15045

**Questions:**

1. Why is cubic activation used in BERT experiments? Is it related to the differentiable loss assumption?
2. The finite difference estimation for $L, \beta$, etc is this done after some n steps during training? Or is it done only once?
3. The theoretical analysis has a strong assumption on convex losses, but practically, we usually deal with highly non-convex, non-smooth objectives. How does the observation of smoothness impacting sparsity and stability get affected?

---

### Official Review · Reviewer_DBy8 · 2024-11-03

**Soundness:** 1
**Presentation:** 2
**Contribution:** 2
**Rating:** 1
**Confidence:** 4

**Summary:**

This paper investigates the impact of MoE sparsity in the trainability of the MoE. Specifically, the paper tries to establish a relationship between the number of experts activated during routing and the stability of the SGD updates. Based on their investigation, they propose a new routing mechanism.

**Strengths:**

- The question this paper raises is interesting and important. There is generally a lack of theoretical analysis in MoE literature and thus a lack of concrete understanding on the facets of instabilities in sparse/discrete MoEs.

**Weaknesses:**

- Inconsistent use of notation: At L182, $\mathcal{B}$ is considered a batch, and in L187, it is a "dataset" which includes many batches. Is it still a batch in L187? If so, there would be no randomness in Eq. (1) w.r.t. $\mathcal{U}$. However, in Eq. (1), the notation $\mathcal{U}(\mathcal{B})$ implies it is a batch based on your notation from L182. In L214, you use $\mathcal{B}$ again, but here it denotes a norm ball.
- It's hard to believe Proposition 1. While Proposition 1 claims that the router has 0 gradients when Top-1 routing is preformed, Top-1 routing has been shown to work in practice, and the router does change throughout training. How do the authors reconcile this discrepancy?
- Crucially, I believe the experimental setup is a bit weird and incorrect. In the experiments, the authors enforce expert specialization prior to training the MoE. This is highly non-standard and significantly deviates from practice. In practice, the MoE is initialized and the whole MoE is trained from scratch. The point is that simply through gradient descent, the router learns non-trivial routing and the experts develop specialization during training. Given this significant deviation from practice, it becomes difficult to appreciate the empirical evaluations and their implications.
- Not enough details on the experiment setup - e.g. for the transformer model, how was pre-training +fine-tuning performed. Were these two training steps done prior to training the MoE? Did you just swap out the MLP in the original non-MoE BERT model after the fine-tuning phase?
- No code.

Other points
- Equation at L137 for $\mathcal{F}\_{i}(\Theta_i; \mathbf{x})$ should be $\mathcal{F}\_{i}(\Theta_i; \mathbf{z}_i)$
- In your discourse on L-Lipschitz in L169, you describe the following: "L-Lipschitz $\mathcal{L}$" implies "bounded gradient of $\mathcal{L}$" implies "actual definition of L-Lipschitz". Technically, this is incorrect.

**Questions:**

- Intuitively, I'm not sure how the $\frac{L^{2}}{\beta}$ metric is an indicator of "how rapidly the loss function can change". The $L^2$ term makes sense, but the inverse relation to $\beta$ doesn't seem to make sense in this context. You purport to measure gradient smoothness with this metric, but doesn't $\beta$, the Lipschitz constant of the gradient of the loss, suffice?
- L712 says $g_{i,j}$ is bounded according to Assumption 1, but I don't see how Assumption 1 implies boundedness of $g_{i,j}$.

---

### Meta-Review · Area_Chair_uSUN · 2024-12-21

**Metareview:**

In the paper, the authors investigate the interplay between expert sparsity and training stability in Mixture-of-Experts (MoE) models. The authors theoretically and empirically show that activating more experts enhances gradient smoothness and stabilizes the training. Finally, they introduce a new routing mechanism that employs independent router heads and a soft top-$K$ expert activation strategy via sampling without replacement. It achieves smooth gradient landscapes while preserving expert sparsity.

There are several weaknesses in the current paper: (1) The assumptions in the paper and key results, such as Theorem 1, are questionable and contradictory. For instance, Top-K routers are known to be discontinuous but the authors assume Lipschitz continuity as the theoretical foundation for the entire framework. (2) The presentation of the paper is quite poor (e.g., notations are inconsistent). (3) The experiments are poor and not convincing.

Given the above weaknesses, I recommend rejecting the paper at the current stage. The authors are encouraged to incorporate the feedback and suggestions of the reviewers into the revision of their manuscript.

**Additional Comments On Reviewer Discussion:**

Please refer to the meta-review.

---

### Decision · Program_Chairs · 2025-01-22

Reject